# Path Tracking of a 4WIS–4WID Agricultural Machinery Based on Variable Look-Ahead Distance

Lijun Xu, Yankun Yang, Qinhan Chen, Fengcheng Fu, Bihang Yang and Lijian Yao *

College of Optical, Mechanical and Electrical Engineering, Zhejiang A&F University, Hangzhou 311300, China
* Correspondence: ljyao@zafu.edu.cn; Tel.: +86-0571-6379-3974

**Abstract:** Aiming to solve the problem of the low path-tracking accuracy of mobile robots in agricultural environments, the authors of this paper propose a path-tracking method for agricultural machinery based on variable look-ahead distance. A kinematic model of the four wheel independent steering–four wheel independent drive (4WIS–4WID) structure based on pure pursuit was constructed to obtain the functional equation of the current position and the four-wheel steering angle. The fuzzy controller, which takes the lateral deviation and heading deviation as input and the look-ahead distance in a pure pursuit model as output, was designed to obtain the look-ahead distance that changes dynamically with the deviation of mobile agricultural machinery. The path-tracking performance of 4WIS–4WID agricultural machinery in three scenarios (1 m, $-90°$; 1 m, $0°$; and 0 m, $90°$) with different initial deviations was tested using a pure pursuit model based on a variable look-ahead distance. The obtained field test results showed an average deviation of 19.7 cm, an average tracking time of 5.1 s, an average stability distance of 203.9 cm, and a steady state deviation of 3.1 cm. The results showed that the proposed method presents a significant path-tracking performance advantage over a fixed look-ahead distance pure tracking model and can be a reference for high-quality path-tracking methods in automatic navigation research.

**Keywords:** look-ahead distance; pure pursuit model; fuzzy control; path tracking





## 1. Introduction

Automatic navigation technology for intelligent agricultural machinery is one of the most important support technologies for modern agricultural equipment [1] that guarantees a high level of agricultural production. Path tracking is a key technology of automatic navigation. A pure pursuit model is a geometric method used to simulate a driver's driving behavior in which the steering radius is calculated in real time by its own posture information and a preset look-ahead distance without relying on a vehicle dynamics model [2,3]. It is widely used in unmanned agricultural machinery path tracking and path control.

In narrow environments, such as greenhouses, intelligent agricultural machinery needs to have a good path-tracking ability [4]. The value change of look-ahead distance, as the only parameter that can be adjusted in a pure pursuit model, has a great influence on the path-tracking ability of agricultural machinery [5]. A number of researchers have designed experiments and tests to improve the accuracy of path tracking. When the look-ahead distance is equal to the width of the wheel, the obtained path-tracking accuracy is limited [6]. With a tracking algorithm based on an optimal target point, although the accuracy of path tracking could be improved, it is too complex for environments with high real-time requirements such as greenhouses [7]. A pure pursuit model combined with GPS used to obtain the parameters and positions of a vehicle, as well as the look-ahead distance through simulation, could improve tracking accuracy but requires the simulation of different speeds and too-complicated operation [8]. In research on fixed look-ahead distance, researchers have found that a dynamic look-ahead distance performs better in path tracking. Although it considered the influence of speed, a pure pursuit model with

a PID algorithm used to change the look-ahead distance was not found to be efficient [9]. Based on Kalman filtering and a pure tracking model, Zhang et al. [10] used the IATE optimization criterion to simulate the optimal look-ahead distance. Although the path-tracking ability was improved in their study, the application of various complex functions is needed, which is not conducive to the operation of agricultural machinery. A PI-based path controller that changes the look-ahead distance without considering the influence of speed is thus not dynamic [11]. An adaptive method that changes the look-ahead distance could improve the pure pursuit algorithm, but its algorithm requires extensive analysis that is too complex and not convenient for the operation of farm machinery [12]. Determining the look-ahead distance of a pure tracking model with the fuzzy adaptive method requires the construction of a self-adjustment function and complicated operation [13].

During travel, the determination of look-ahead distance is strongly related to each driver's experience, so researchers have attempted to find more intelligent methods to determine the optimal look-ahead distance. An adaptive, look-ahead distance, pure pursuit lateral controller that uses a greedy algorithm to improve the tracking accuracy was proposed, but it is not suitable for complex terrain situations [14]. A path-tracking model based on a fuzzy controller that could provide the optimal steering angle was constructed, but the vehicle structure and kinematics model were different from our test prototype [15]. Hu determined the optimal look-ahead distance according to the curvature of the reference path and the current speed [16]. An improved pure pursuit algorithm using AGV to predict the trajectory and turning speed could improve the accuracy of path pursuit, but the adopted technology is not suitable for agricultural conditions [17]. With an extended Kalman filter, the method could decompose speed into Cartesian components and improve path-tracking accuracy, but the amount of calculations was large and unsuitable for the operation of agricultural machinery [18]. When using pure pursuit and proportional integral speed to improve the effect of path planning, vehicles will not collide with obstacles, but this is not conducive to farming in farmland [19]. A lateral controller based on LPV-MPC was shown to have some trouble handling uncertainties, while nonlinear active disturbance rejection control was found to perform slightly worse regarding path tracking but had strong robustness [20].

In conclusion, researchers have performed much research on the look-ahead distance of pure pursuit models, but the above-mentioned methods require many experiments to determine their key parameters, and the real-time performance of control algorithms needs to be improved. Furthermore, few related studies on the adoption of pure pursuit on 4WIS–4WID vehicle have been found. Fuzzy control is an intelligent control algorithm with artificial experience that is robust, fault-tolerant, and suitable for nonlinear and time-varying system control [21,22]. Intelligent agricultural machinery field path-tracking requires various inputs and outputs that are difficult to describe with a simple linear model. Therefore, the authors of this paper propose an improved pure pursuit model using the fuzzy control method to calculate the optimal look-ahead distance in real time to obtain better agricultural machinery path-tracking quality and proved its effectiveness through real vehicle tests. This method can provide new ideas for intelligent agricultural machinery path-tracking research.

## 2. Materials and Methods

### 2.1. Test Prototype

The test prototype was a 4-wheel independent steering, 4-wheel independent drive mechanical structure (4WIS–4WID), as shown in Figure 1. The body size of the test prototype was 120 cm × 70 cm × 70 cm (length, width, and height, respectively), with a left and right wheel tread of 54 cm and a wheelbase of 104 cm. With a lithium battery (24 V, 20 Ah) as the power source, STM32F103ZET6 (32 bit, 72 MHz) was used as the main controller to receive the attitude information of the vehicle position (x, y) and orientation information $\theta$ sent by UWB (ultra-wideband; positioning error: $\pm 5$ cm) and electronic gyroscope (WT901C; error: $\pm 0.1°$), respectively. The PWM signal controlling the vehicle

steering and movement were generated by the path-tracking model were transmitted to the steering gear and drive motor. The steering of the wheels was independently controlled by four steering gears (DH-03X, 120°, 38 N/M) at [−90° 90°] (setting a positive angle for the wheel counterclockwise and a negative angle for clockwise deflection). The four hub motors were driven by AQMD6015BLS drivers to realize prototype driving. The control process is shown in Figure 2.

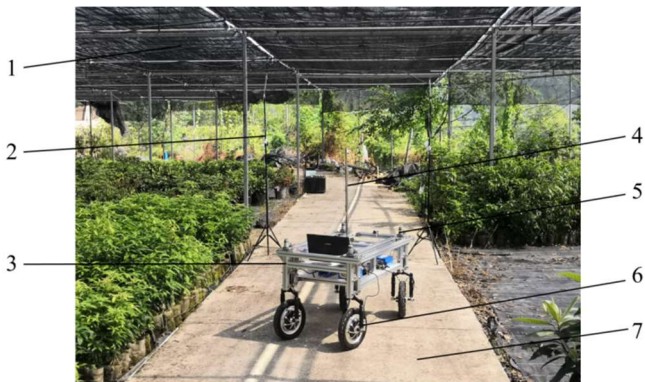

**Figure 1.** Test prototype and test environment. (1) Greenhouse; (2) UWB anchor; (3) test prototype; (4) UWB signal receiver; (5) steering gear; (6) drive motor; (7) test path.

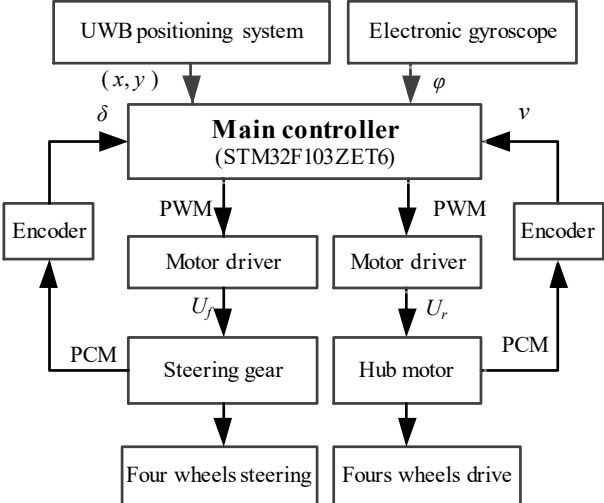

**Figure 2.** System control process.

When tracking the prototype vehicle path, the main controller reads the angle data of the position coordinates (x, y) and the deviation data $\varphi$ provided by the UWB wireless positioning system and the electronic gyroscope, respectively, according to the sampling period and then converts the position and attitude information into the lateral deviation $d$ and heading deviation $\theta$ relative to the desired path. A dual-input and single-output fuzzy controller was designed to construct a nonlinear transfer model between body deviation and the look-ahead distance $LD$. The lateral deviation $d$, heading deviation $\theta$, and look-ahead distance $LD$ are input into the pure pursuit model to calculate the steering radius $R$ of the prototype under the current state, and then the rotation angle of each wheel $\delta_1$, $\delta_2$, $\delta_3$, and $\delta_4$ and the driving speed of the prototype wheel $v_1$, $v_2$, $v_3$, and $v_4$ are calculated by the four-wheel, Ackermann low-speed steering model. The main controller converts the rotation angle $\delta$ and speed $v$ of each wheel into PWM signals that act as input of the steering gears and the wheel motor to realize driving and steering so that the path-tracking control of the prototype vehicle can be accomplished. The fuzzy control process is shown in Figure 3.

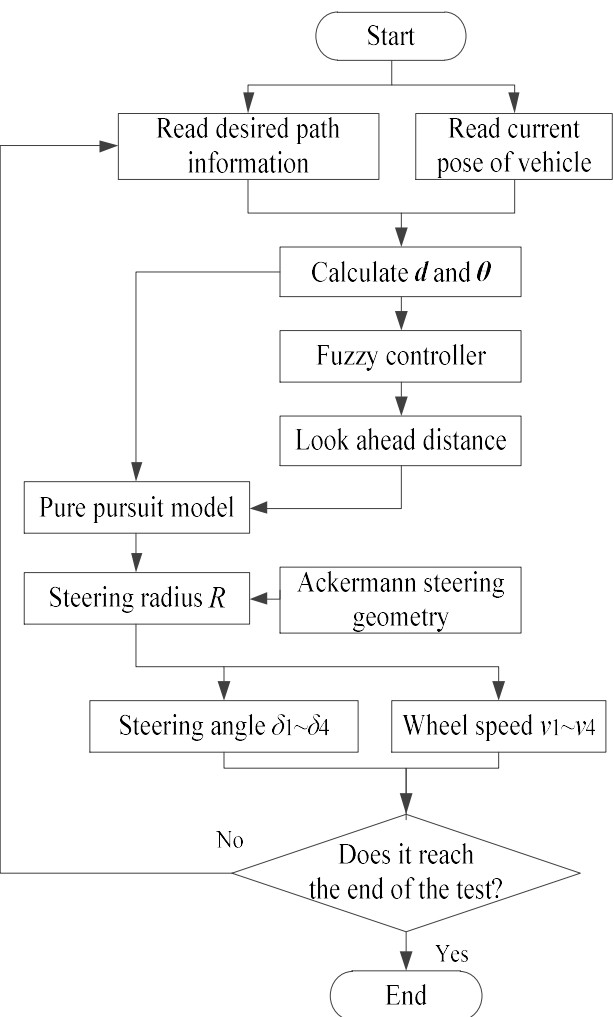

**Figure 3.** Fuzzy control process.

*2.2. Path-Tracking Method Design*

2.2.1. Pure Pursuit Model

The pure pursuit model was set as a lateral motion control algorithm for vehicles that simulates the driver's experience. In Figure 4, X is the geometric center of the 4WIS–4WID prototype vehicle, line $P_1P_2$ is the desired path, and the angle between vehicle forward direction and line $P_1P_2$ is the heading deviation $\theta$. Crossing the center point X, line XN is drawn perpendicular to line $P_1P_2$ and point N is the intersection points, so XN is the lateral deviation $d$ of the vehicle. If we set the length of MN equal to the look-ahead distance $LD$, then point M is the look-ahead point of the prototype vehicle and XM is the look-ahead straight line. For the pure pursuit model, the steering center O of the prototype vehicle must be at the intersection of lines GO and XO. GO is the vertical bisection line of the look-ahead line XM, and XO is the extension line of the vehicle's lateral center. That is, OX is the steering radius $R$ of the prototype vehicle under the current posture state.

According to the geometric relationship in Figure 4, the steering radius $R$ by Equation (1) is:

$$R = \frac{LD^2 + d^2}{2LD \sin \theta + 2d \cos \theta} \tag{1}$$

where

$R$—steering radius, m;
$d$ = the lateral deviation, m;
$\theta$ = the heading deviation, °.

From Equation (1), when the deviation is fixed, look-ahead distance *LD* is the only parameter to determine the rotation radius and there is a positive correlation between *LD* and *R*, i.e., the greater the *LD*, the greater the value of *R* and vice versa.

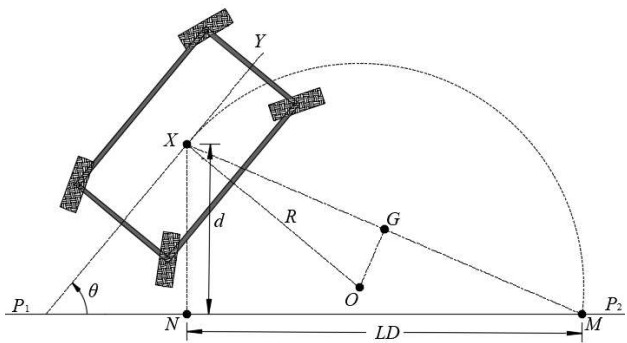

**Figure 4.** Pure pursuit model.

### 2.2.2. Kinematics Equation of the Vehicle

Different from traditional rear wheel-driven vehicles, to control a 4WIS–4WID vehicle, steering angle $\delta_1 \sim \delta_4$ and speed $v_1 \sim v_4$ for each of the four wheels need to be considered, thus leading to an even more complicated kinematics model as Figure 5.

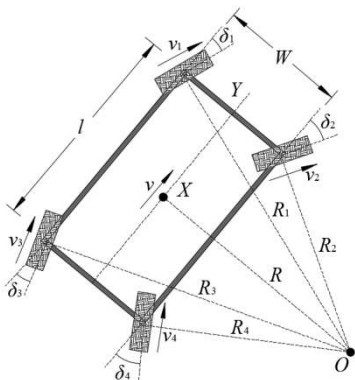

**Figure 5.** Kinematics model of the vehicle.

According to the Ackermann steering principle of Equation (2),

$$\cot \delta_1 - \cot \delta_2 = \frac{2W}{l} \tag{2}$$

where:

$l$ = wheelbase, m;
$W$ = left and right wheel tread, m;
$\delta_1$ = steering angle of the front outer wheel, °;
$\delta_2$ = steering angle of the front inner wheel, °.

Since $2W/l > 0$, the value of $\delta_1$ is less than $\delta_2$. In the simplified steering model, the front and rear wheels on the same side have unique steering angles but opposite steering directions, i.e., $\delta_1 = -\delta_3$, $\delta_2 = -\delta_4$, as in Equation (3).

$$\begin{cases} \delta_1 = -\delta_3 = \arctan\frac{l}{2R+W} \\ \delta_2 = -\delta_4 = \arctan\frac{l}{2R-W} \end{cases} \tag{3}$$

where:

$l$ = wheelbase, m;
$W$ = left and right wheel tread, m.

The wheel is required to turn clockwise in a positive direction and counter-clockwise in a negative direction.

Thus, rotation radii for the four wheels are determined by Equation (4).

$$\begin{cases} R_1 = R_3 = \frac{2R+W}{2\cos\delta_1} \\ R_2 = R_4 = \frac{2R-W}{2\cos\delta_2} \end{cases} \tag{4}$$

To ensure that the four wheels are always pure rolling with the ground during steering, the rotation speeds of the four wheels should be in accordance, and the speeds of the four wheels $v_1$–$v_4$ are listed as Equation (5).

$$\begin{cases} v_1 = v_3 = \frac{vR_1}{R} \\ v_2 = v_4 = \frac{vR_2}{R} \end{cases} \tag{5}$$

where: $v$ = driving speed of the prototype vehicle, m/s.

Thus, the kinematic control model of the prototype vehicle could be constructed. In the model, deviation ($d$, $\theta$) can be transferred into the steering angle $\delta_1{\sim}\delta_4$ and wheel speed $v_1{\sim}v_4$.

In the pure pursuit model, the forward distance $LD$ has a significant impact on the path-tracking quality. A smaller forward distance quickly converges the prototype to the desired path (desired path) but also reduces the stability of the path tracking; a larger forward distance enables the prototype to smoothly converge to the desired path, but it takes a longer time and passes over a longer distance. The experience shows that when the deviation of the prototype vehicle is large, a small look-ahead distance should be used to quickly adjust the vehicle position and converge to the desired path; when the deviation of the prototype vehicle is small, a large look-ahead distance is used to avoid the oscillation caused by over-sensitive system adjustment. Currently, there is no mature mathematical function used to determine the look-ahead distance for vehicle with a 4WIS–4WID structure. A fuzzy controller was designed to dynamically determine the look-ahead distance $LD$ using the driver's driving experience as the control rule.

### 2.2.3. Design of Fuzzy Controller

(1) Membership Function

The lateral deviation $d$ and the heading deviation $\theta$ of the test prototype and the desired path were taken as the input variables of the fuzzy controller. The domain of $d$ was [−1.2 m 1.2 m], and the domain of $\theta$ was [−90° 90°]. The look-ahead distance $LD$ was regarded as the output variable, and the domain was [1 m 2.2 m]. The positive and negative signs of each variable were defined as follows: when the vehicle is located on the left forward side of the desired path, $d$ is positive, and when the vehicle is located on the right forward side of the desired path, $d$ is negative. The heading deviation $\theta$ is positive when rotated counterclockwise and negative when rotated clockwise; the steering angle is positive when the front wheels turns left and negative when the vehicle turns right. Comprehensively considering the precision of system control and the flexibility of optimization, in the fuzzy controller, the input and output variables were quantified into 5 fuzzy subsets. Different from a traditional fuzzy controller, we adopted non-uniform quantization scale instead of a uniform quantization scale in membership function. That is, a large-scale quantitative level is used to obtain good control stability when vehicle deviation is large. When the deviation is small and the vehicle is close to the desired path, the high resolution (small scale) quantitative level is used to realize the more precise adjustment of the vehicle position and posture. A triangle membership function is used for each input and output variable. The basic information of each input and output variable is shown in Figure 6.

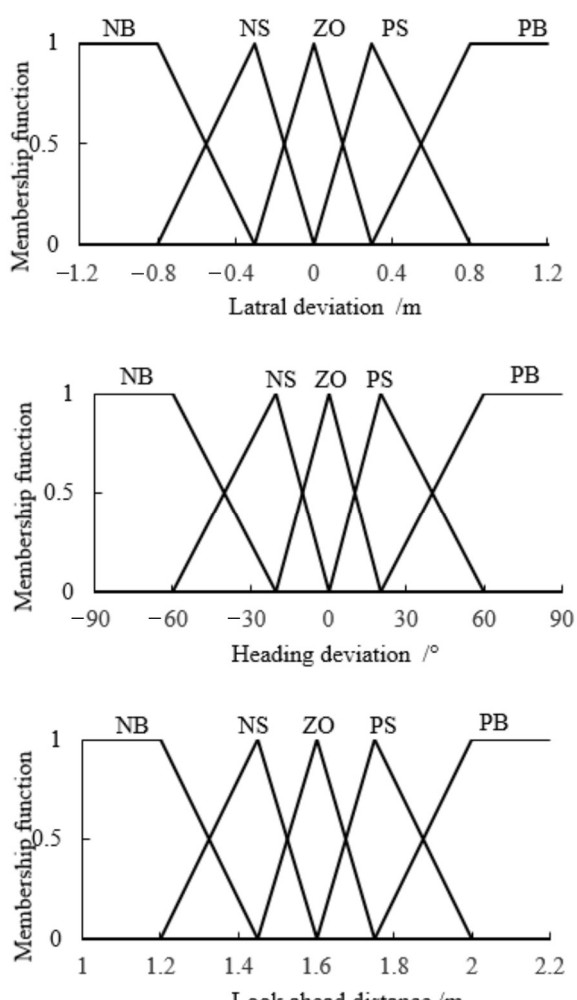

**Figure 6.** Membership functions.

(2)    Control Rules

Based on the experience of experts and drivers, the control rules should be in accordance with following principles. When the deviation is very large, a small look-ahead distance should be selected to quickly converge to the desired path to improve the efficiency of path tracking. In contrast, when the deviation is small, the look-ahead distance should be appropriately increased to prevent oscillation during path tracking, thus increasing the stationarity of the prototype vehicle. Through repeated simulation and debugging, 25 control rules were determined, as shown in Table 1. In Table 1, the upper left and lower right corners show that the vehicle is located in a position with large lateral deviation *d* and heading deviation *θ*, so smaller look-ahead distance scales NB and NS are used; nearby, the middle and secondary diagonals (the line from the lower left to the upper right corner in the table) are the locations where the deviation of the vehicle (*d*, *θ*) is small, and the greater look-ahead distances PB and PS are used.

**Table 1.** Control rules.

| LD | | θ | | | | |
|---|---|---|---|---|---|---|
| | | **NB** | **NS** | **ZO** | **PS** | **PB** |
| | NB | NB | NB | NS | ZO | PS |
| | NS | NS | NS | ZO | PS | PS |
| d | ZO | ZO | PS | PB | PS | ZO |
| | PS | PS | PS | ZO | NS | NS |
| | PB | PS | ZO | NS | NB | NB |

## 3. Results

### 3.1. Path-Tracking Test Design

To verify the effectiveness of the method in path-tracking quality improvement, a real vehicle test was conducted in Guan-Tang Greenhouse of Zhejiang Agricultural and Forestry University located in Zhejiang province, south east of P.R. China from April to May 2022. The UWB anchors were placed at four vertices of a 2 m wide and 10 m long rectangle for receiving and transmitting position signals. The frequency of data and signal sampling was 5 Hz, the vehicle speed $v$ was set at 0.6 m/s, and the length of the straight line for the path-tracking test was 10 m, as shown in Figure 7.

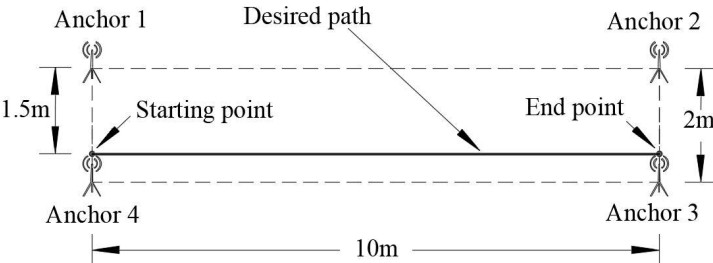

**Figure 7.** Layout of test site.

Considering the frequent occurrence of the right angle steering of agricultural machinery in actual greenhouse operation, three common deviation scenarios (1 m, −90°; 1 m, 0°; and 0 m, 90°) were designed as the initial states of the prototype. The former data in brackets comprise the lateral deviation d, and the latter comprise the heading deviation θ. The pure tracking model with a fixed look-ahead distance was contrast-tested, and the fixed look-ahead distance used in the comparison test was set to 1.5 m.

The average deviation, maximum deviation, stability distance, stability time, and steady-state deviation were used as the evaluation indicators of path-tracking quality. The average deviation refers to the average of the lateral deviation of all sampling points throughout the whole path tracking. Stability distance refers to the distance that the prototype vehicle travelled from the initial state to the stable state, i.e., the distance from converging to the forward direction when the lateral deviation was less than 0.1 m. The maximum deviation refers to the maximum lateral deviation of the actual path relative to the desired path. Stability time refers to the time elapsed from the initial state to the steady state when lateral deviation was less than 0.1 m. Steady state deviation refers to the mean of the lateral deviation of all data points after the prototype vehicle reached the steady state.

### 3.2. Test Results

The tracks of the prototype vehicle path tracking with three different initial deviations are shown in Figure 8. When obtaining the track, the starting point of the desired path was taken as the origin. Ldynamic is the tracking track of the variable look-ahead distance in the experimental group, and Lfixed is the tracking track of the fixed look-ahead distance in

the control group. To better compare the test results, we locally enlarged the trajectories in the initial stage of path tracking (0–4 m) as shown in Figure 9.

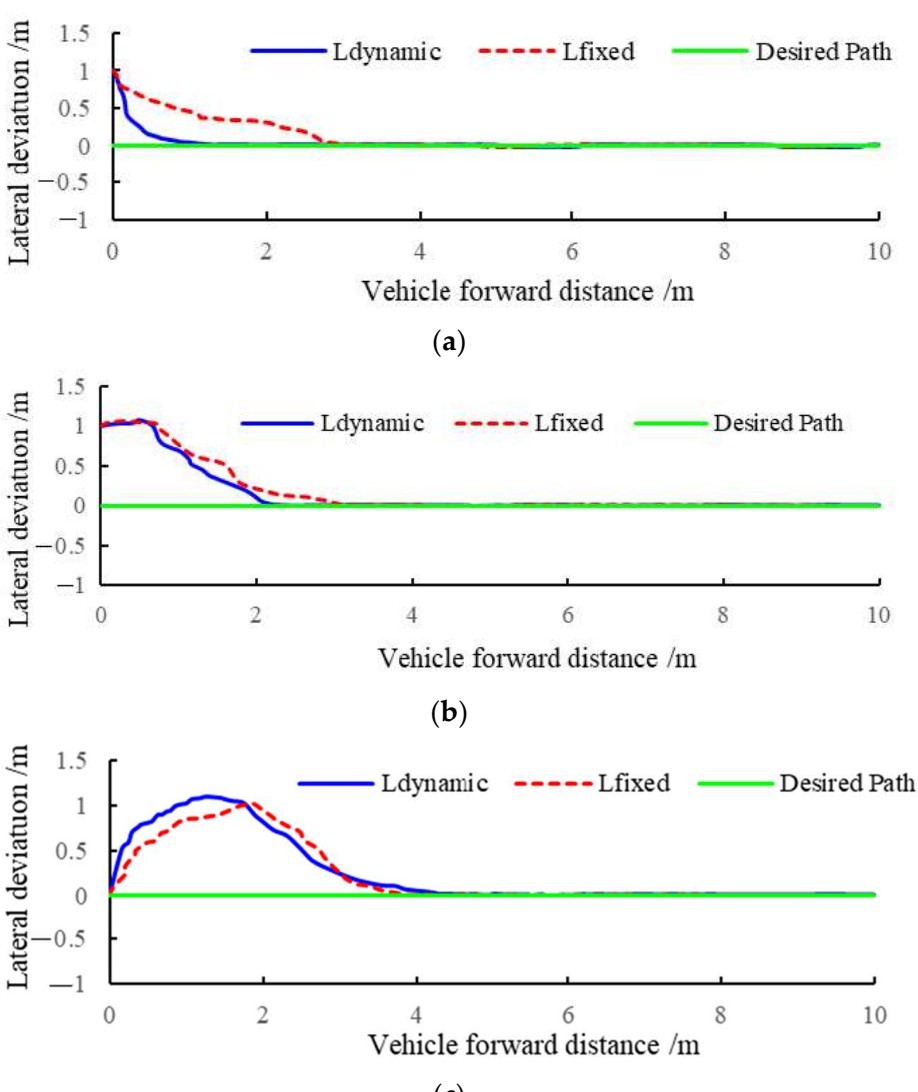

**Figure 8.** Linear tracking track in different initial states. (**a**) Initial state (1 m, −90°) tracking track; (**b**) initial state (1 m, 0°) tracking track; (**c**) initial state (0 m, 90°) tracking track.

The further statistical analysis of the data shown Figure 8 was performed, as shown in Table 2.

**Table 2.** Statistics of the linear tracking test.

| Initial States | Look-Ahead Distance (m) | Average Deviation (cm) | Stability Distance (cm) | Maximum Deviation (cm) | Stability Time (s) | Steady-State Deviation (cm) | Standard Deviation of the Steady-State Deviation (cm) |
|---|---|---|---|---|---|---|---|
| (1 m, −90°) | Ldynamic | 12.5 | 64.6 | 100.0 | 3.2 | 2.0 | 1.3 |
| | Lfixed | 17.5 | 274.7 | 100.0 | 5.6 | 3.4 | 2.1 |
| (1 m, 0°) | Ldynamic | 20.8 | 204.9 | 106.8 | 4.2 | 3.4 | 0.9 |
| | Lfixed | 21.7 | 275.5 | 107.9 | 5.2 | 4.4 | 1.5 |
| (0 m, 90°) | Ldynamic | 25.7 | 342.3 | 100.9 | 7.8 | 3.8 | 1.6 |
| | Lfixed | 34.2 | 372.9 | 109.1 | 8.4 | 4.6 | 1.9 |

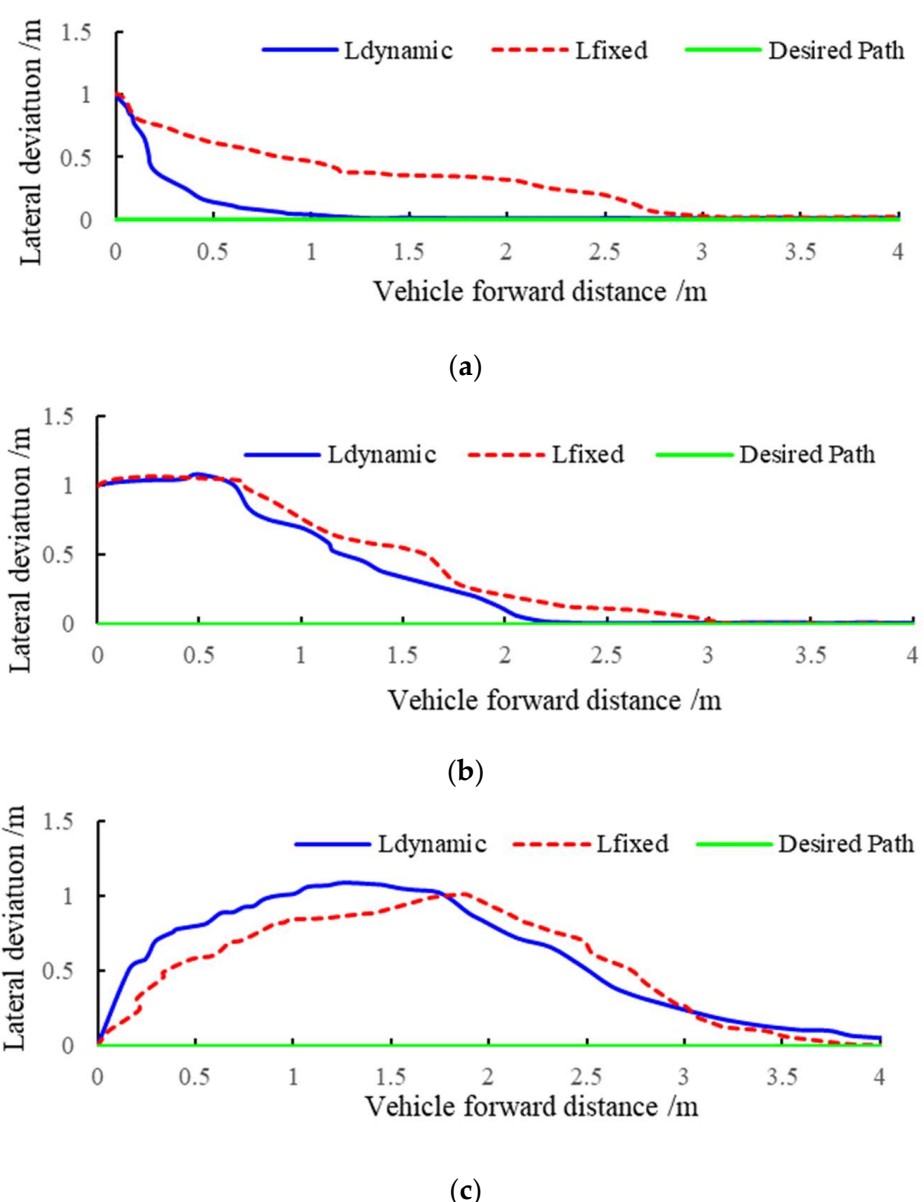

**Figure 9.** Locally enlarged trajectories of Figure 8. (**a**) Locally enlarged trajectories of Figure 8a; (**b**) locally enlarged trajectories of Figure 8b; (**c**) locally enlarged trajectories of Figure 8c.

## 4. Discussion

### 4.1. Path-Tracking Accuracy

The accuracy of path tracking was evaluated with three indexes: mean deviation, maximum deviation, and steady-state deviation. As shown in Figure 8 and Table 2, the average deviations of path tracking for the variable look-ahead distance for three different scenarios (three initial states) were 12.5 cm, 20.8 cm and 25.7 cm, and the average value of deviation of the three scenarios was 19.7 cm. The steady-state deviations were 2.0 cm, 3.4 cm, and 3.8 cm, respectively, and the average value of the three scenarios was 3.1 cm. The maximum deviations were 100 cm, 106.8 cm, and 100.9 cm, respectively. The above-mentioned values were less than the corresponding value of the fixed look-ahead distance in the control group. These results show that the path-tracking accuracy of the presented method was higher than that of the traditional pure tracking model with a fixed look-ahead distance. This is because when the initial deviation was large, the fuzzy controller provided a small look-ahead distance so that the lateral deviation of the vehicle body was quickly reduced, while when the vehicle body had a large look-ahead distance, it adopted a small

deviation that effectively reduced the over adjustment of the lateral deviation control. Thus, selecting the appropriate look-ahead distance in real time according to the current deviation state could effectively control and stably control the actual tracking path of the prototype vehicle closer to the desired path.

*4.2. Convergence Rapidity*

The stability distance and the stability time were used to evaluate the convergence speed. According to Figure 8 and Table 2, with variable look-ahead distance, the stability distances for the three different initial states were 64.6 cm, 204.9 cm, and 342.3 cm, respectively, and the average value of stability distances was 203.9 cm. Stability times were 3.2 s, 4.2 s, and 7.8 s, respectively, and the average value was 5.1 s. In contrast, with the fixed look-ahead distance, stability distances for the three different initial states were 274.7 cm, 275.5 cm, and 372.9 cm, respectively. Stability times were 5.6 s, 5.2 s, and 8.4 s, respectively. The method used in this paper can be used to achieve a more rapid convergence to the desired path and has a more sensitive response. In the early period of path tracking, the small look-ahead distance shortens the "rise time" in the control process, thus shortening the time required for the prototype vehicle to converge from the initial state to the steady state. In real agricultural production situations, the characteristics of rapid convergence can reduce the idle time to obtain a higher operational efficiency.

*4.3. Path-Tracking Stability*

The standard deviation of the steady-state deviation is used as an index to evaluate the stability of path tracking. This index can reflect the transverse deviation dispersion degree of a sample car after entering the stable state and can thus reflect the stability degree of the sample car when driving at this stage. As shown in Figure 8 and Table 2, the standard deviation of the steady state deviations of the present method under three different initial deviations in the test were 1.3 cm, 0.9 cm and 1.6 cm, respectively, while the standard deviations under the fixed look-ahead distance were 2.1 cm, 1.5 cm, and 1.9 cm, respectively. The experimental data showed that the dispersion of the lateral deviation of the prototype vehicle was significantly less than that of the control group with the fixed look-ahead distance, so the path-tracking model with a variable forward distance had a better performance than the fixed look-ahead distance. In the small-deviation situation, the fuzzy controller was used to dynamically select a large look-ahead distance for path tracking, which could preferentially reduce over-regulation in the control process, avoid oscillation, and improve the stability of path tracking.

**5. Conclusions**

(1) To further improve the path-tracking quality of automatic navigation in agricultural machinery with a 4WIS–4WID structure, a pure pursuit model based on a variable look-ahead distance was adopted. A fuzzy controller was designed, with lateral deviation and heading deviation as the input and look-ahead distance as the output, to obtain dynamic variable look-ahead distances. A real vehicle path-tracking test was implemented in a real agricultural environment to validate the effectiveness of the algorithm in path tracking.

(2) The dynamic adjustment of the variable look-ahead distance according to the deviation of the vehicle matches the driving habits of experienced drivers. The non-uniform membership function quantization method can guarantee the accuracy of path tracking and consider the speed and stability of the path tracking. Compared with the fixed look-ahead distance method, the method presented in this paper improved the performance of average deviation, average steady-state deviation, average steady-state distance, average maximal deviation, and the average stability time by 19.6%, 24.4%, 33.7%, 2.9% and 20.3%, respectively, according to the comparison test of the test prototype. The path-tracking accuracy, convergence rapidity, and stability were significantly improved compared to those of the traditional fixed look-ahead distance

method. The path-tracking method can be applied to multi-input, nonlinear and time-varying system control.

(3) The quantitative scale design of fuzzy control rules can be further refined. In follow-up research, more fine fuzzy control rules can be used, and the input and output variables can be quantified into more fuzzy subsets to determine a variety of deviation states with more accurate look-ahead distances, further improving the path-tracking accuracy of automatic navigation technology for agricultural machinery.

**Author Contributions:** Conceptualization, Y.Y. and F.F.; methodology, Y.Y., F.F. and L.Y., fata curation, B.Y.; software, Y.Y., F.F. and Q.C.; writing—review and editing, Y.Y., L.X. and L.Y. All authors have read and agreed to the published version of the manuscript.

**Funding:** This research was funded by Key R&D Program of Zhejiang, grant number 2022C02042.

**Institutional Review Board Statement:** Not applicable.

**Informed Consent Statement:** Not applicable.

**Data Availability Statement:** Not applicable.

**Conflicts of Interest:** The authors declare no conflict of interest. The funders had no role in the design of the study; in the collection, analyses, or interpretation of data; in the writing of the manuscript; or in the decision to publish the results.

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
