# Peer review of "Path Tracking of a 4WIS–4WID Agricultural Machinery Based on Variable Look-Ahead Distance"

_applsci, doi:10.3390/app12178651_

Round 1

Reviewer 1 Report

The paper is about controlling of 4WIS-4WID wehicle. I would like to have sooner explanation what IS and ID means. 

According to the paper the proposed algorithm has been tested on real vehicle in real environment. I would like to have picture of the vehicle and maybe also of the environment. I found that authors have such picture in their previous papers...

The paper is about interesting topic but the overall feeling is that the topic is quite simple... Maybe there should be more testing and comparing of the proposed algorithm with other solutions. Also the Figure 3 is quite confusing, maybe would be better to split it into two figures.

Author Response

Dear reviewer,

thank you for your valuable comments, we response them point to points, Please see the attachment.

Reviewer 2 Report

This paper proposes a path tracking method for agriculture machinery based on variable look ahead distance. The kinematic model of the 4WIS-4WID structure is used to calculate the steering angle of the mobile robot. To validate the proposed method three scenarios are considered by the author which is based on the pursuit model. The author claims that when compared to fixed look ahead distance the proposed method outperformed and can be used for mobile robot navigation.

Furthermore, to improve the quality of the manuscript, the authors are suggested to incorporate the following comments as a revised version.

1.      Introduction

a.      The introduction part is inadequate, I recommend the author add a few more pieces of content explaining the importance of this research.

b.      In related work the author has concluded that the fuzzy control method provides better agriculture machinery path tracking quality. But reviewer couldn’t find any related work which utilizes a fuzzy-based concept. I recommend the authors add a few more latest related works to this section. It will be better to exploit a tabulation comparing what is lacking in other methods and how the proposed method can perform optimal look ahead distance in real-time.    

2.      Materials and Methods

a.      How the lateral and heading deviation is calculated ?? it will be better to incorporate them in this section.

b.      As per figure 2 the desired path has not been given as an input to the fuzzy controller. From where the desired path is provided?? Lateral d and heading deviation theta are given to the fuzzy controller.

c.      What is the state of the art in this work ?? I recommend the author to focus and highlight while explaining the proposed work.

3.      Results

a.      As said in the abstract I could not find three scenarios to validate the proposed method. I recommend the author justify or incorporate the contents in this section.

b.      In figure 5, the reviewer cloud sees less difference between Ldynamic and Lfixed. How can the author claims the proposed method is better? Justify or incorporate the relevant content in this section.

4.      Conclusion

a.      The conclusion looks like a summary of the work. The conclusion should be extended with more personal valuation by the authors including a take-home message.  I recommend the author rewrite this part.      

Author Response

(The authors gave the same response as above.)

Round 2

Reviewer 2 Report

The author has satisfactorily addressed my previous comments and modified their manuscript accordingly. Hence, I am glad to recommend the present work for publication.